# Fine Structure of the Carbon-Related Blue Luminescence Band in GaN

**Michael A. Reshchikov**

Department of Physics, Virginia Commonwealth University, Richmond, VA 23220, USA; mreshchi@vcu.edu; Tel.: +1-(804)-828-1613

**Abstract:** Photoluminescence studies reveal three $C_N$-related luminescence bands in GaN doped with carbon: the YL1 band at 2.17 eV caused by electron transitions via the $-/0$ level of the $C_N$, the $BL_C$ band at 2.85 eV due to transitions via the $0/+$ level of the $C_N$ and the BL2 band at 3.0 eV attributed to the $C_N H_i$ complex. The $BL_C$ band studied here has the zero-phonon line at 3.17 eV and a phonon-related fine structure at low temperatures. The $0/+$ level of the $C_N$ is found at $0.33 \pm 0.01$ eV above the valence band, in agreement with recent theoretical predictions. These results will help to choose an optimal correction scheme in hybrid functional calculations.

**Keywords:** photoluminescence; point defects; GaN; zero-phonon line





## 1. Introduction

GaN attracts the attention of researchers due to its applications in light-emitting devices and high-power electronics [1]. Point defects that may detrimentally affect the material and device efficiency and longevity are still not well understood. Photoluminescence (*PL*) reveals the properties of the defects, such as the charge state, transition levels and carrier capture coefficients [2,3]. Defects often cause broad *PL* bands due to strong electron–phonon coupling.

However, in some instances, fine structure is observed, including zero-phonon line (ZPL) and phonon replicas. For *PL* caused by electron transitions from the conduction band to the $-/0$ transition level, the ZPL energy is $E_0 = E_g - E_i$, where $E_g$ is the bandgap and $E_i$ is the distance from the $-/0$ level to the valence band, also equal to the acceptor ionization energy if there is no barrier for the hole capture. Thus, observation of the ZPL allows us to find the charge transition level with high precision and compare it with predictions of first-principles calculations [4]. Examples include the $Zn_{Ga}$ and $C_N$ acceptors and the $C_N H_i$ donor in GaN [5–7].

The $C_N$ acceptor causes the yellow luminescence band (YL1) in GaN with a maximum at 2.17 eV and ZPL at 2.59 eV [6]. This *PL* band is commonly observed in unintentionally doped GaN grown by various methods and is especially strong in GaN grown by metalorganic chemical vapor deposition (MOCVD) due to contamination with carbon [2,3,8]. In addition to the YL1 band associated with electron transitions from the conduction band to the $-/0$ transition level of the $C_N$ acceptor, a blue band ($BL_C$) can be observed in GaN, which is caused by electron transitions from an excited state close to the conduction band to the $0/+$ level of the $C_N$ [9].

First-principles calculations using hybrid functionals predict the $0/+$ level of the $C_N$ at 0.26–0.50 eV above the valence band and the *PL* band maximum at $\hbar\omega_{max} \approx 2.7$ eV [9–13]. The uncertainty in the calculated $0/+$ level can be related to different correction schemes in hybrid functional calculations [13]. From the analysis of the $BL_C$ band shape at low temperatures, we estimated that $E_i \approx 0.3 \pm 0.1$ eV and $\hbar\omega_{max} \approx 2.85 \pm 0.1$ eV [9].

However, the ZPL has not been found, which limited the accuracy of the $E_i$. The $0/+$ level of the $C_N$ defect was also detected in deep-level transient spectroscopy (DLTS)

experiments [14]. Narita et al. [14] estimated $E_i$ = 0.29 eV at $T \approx 140$ K. However, the ionization energy measured by DLTS may be inaccurate because of the narrow rate window, temperature-dependent ionization energy, and a barrier for the hole capture [15,16]. In particular, the difference between the activation energy (measured by DLTS) and the ionization energy (associated with the defect level) can reach 0.4 eV for $C_N$ [16]. In this work, the ZPL and phonon-related fine structure have been found for the $BL_C$ band in C-doped GaN.

## 2. Materials and Methods

Carbon-doped 1 μm-thick GaN layers were grown on top of 1.7 μm-thick undoped GaN on sapphire substrates by MOCVD at the Institute für Physik, Magdeburg, Germany [17]. The high crystal quality of the samples was confirmed by scanning transmission electron microscopy measurements [17]. The concentration of carbon, as determined from secondary-ion mass-spectrometry analysis, is $8 \times 10^{17}$–$7 \times 10^{18}$ cm$^{-3}$, and the samples are high-resistivity. The properties of three GaN:C samples studied in this work are similar to the properties of sample MD91 studied in Ref. [9] and samples of series C in Ref. [17], where other details can be found. The samples were annealed in a nitrogen ambient at $T = 800$ °C. The annealing is necessary to dissociate the $C_N H_i$ complexes that cause a strong BL2 band with a maximum at 3.0 eV [7].

Steady-state *PL* was excited with a HeCd laser and detected by a cooled photomultiplier tube. Other details of *PL* experiments can be found elsewhere [3,9].

## 3. Results

Figure 1 shows *PL* spectra measured at 18 and 130 K at relatively low excitation intensity ($P_{exc} \approx 0.1$ W/cm$^2$). In the as-grown sample, the YL1 and BL2 bands, attributed to the $C_N$ acceptor and $C_N H_i$ donor, respectively, are very strong at $T = 18$ K [7,9]. To deconvolute the *PL* spectra, we simulated shapes of the broad *PL* bands with the following expression [18].

$$I^{PL}(\hbar\omega) = I^{PL}(\hbar\omega_{\max}) \exp\left[-2S_e\left(\sqrt{\frac{E_0^* - \hbar\omega + \Delta}{d_{FC}^g}} - 1\right)^2\right] \tag{1}$$

Here, $S_e$ is the Huang–Rhys factor, $E_0^* = E_0 + 0.5\,\hbar\Omega_e$, $E_0$ is the ZPL energy, $d^g_{FC} = E_0^* - \hbar\omega_{\max}$ is the Frank–Condon shift, $\hbar\Omega_e$ is the effective phonon mode energy, and $\hbar\omega$ and $\hbar\omega_{\max}$ are the photon energy and *PL* band maximum, respectively. The $\Delta$ is a possible minor shift of the *PL* band maximum due to reasons, such as strain in GaN layers grown on foreign substrates. The shapes of *PL* bands are reproducible in different samples, and the parameters used in this work are the same as those published before [3].

With increasing temperature, the BL2 band is quenched above 100 K with the activation energy of 0.15 eV and completely disappears at 130 K. The $BL_C$ band with a maximum at 2.85 eV, well-resolved at 130 K, is attributed to transitions via the 0/+ level of the $C_N$ defect [9]. It is difficult to study the $BL_C$ band in as-grown GaN:C at low temperatures because of the much stronger BL2 band. However, after annealing in $N_2$ ambient at 800 °C, the BL2 band disappears due to the removal of hydrogen from the sample [7], and the $BL_C$ band can be observed at 18 K (Figure 1).

The $BL_C$ band's shape and fine structure can be better resolved at high excitation intensity when the YL1 band is saturated due to its much longer lifetime. In particular, the decay of the YL1 band in the studied samples is nonexponential at $T = 18$ K, and it persists for milliseconds, whereas the *PL* lifetime of the $BL_C$ band is about 1 ns [3]. Figure 2 shows the *PL* spectrum obtained with a focused laser beam ($P_{exc} \approx 100$ W/cm$^2$).

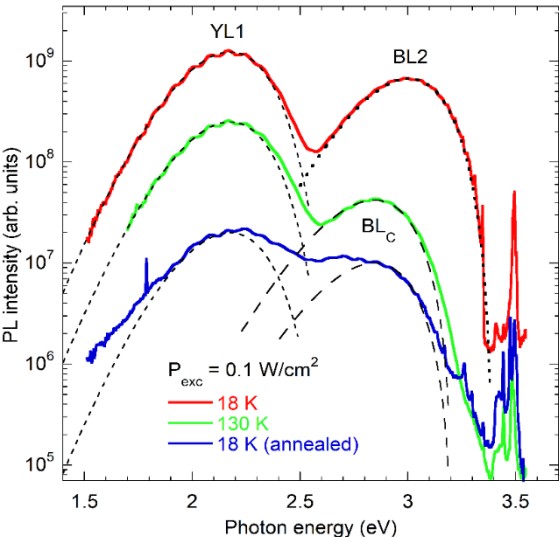

**Figure 1.** PL spectra at 18 and 130 K for as-grown GaN:C and at 18 K after annealing at $T_{ann}$ = 800 °C. The BL2 band is completely quenched at 130 K, revealing the $BL_C$ band. In the annealed sample, the BL2 band is not found even at 18 K. The dashed lines are calculated using Equation (1) with the following parameters: $S_e$ = 7.7, $E_0^*$ = 2.67 eV and $d^g_{FC}$ = 0.50 eV (YL1 band); $S_e$ = 4.4, $E_0^*$ = 3.38 eV and $d^g_{FC}$ = 0.38 eV (BL2 band); and $S_e$ = 3.5, $E_0^*$ = 3.2 eV and $d^g_{FC}$ = 0.35 eV ($BL_C$ band). $\Delta$ = 0.01 eV. A sharp line at 1.79 eV is caused by an internal transition in $Cr^{3+}$ ions in the sapphire substrate, which becomes noticeable when the *PL* signal from GaN is weak.

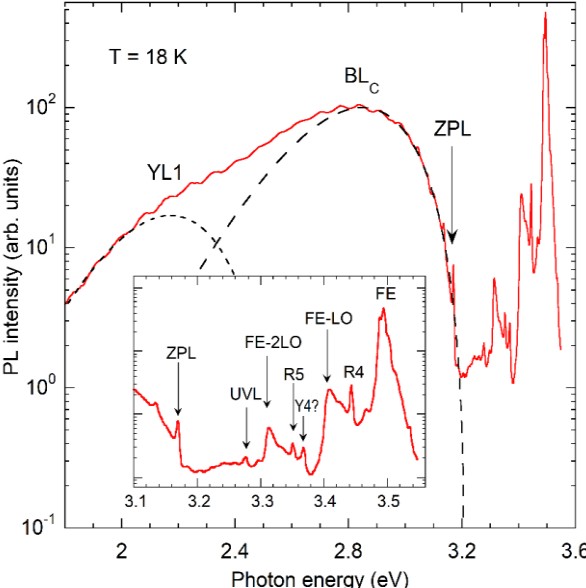

**Figure 2.** *PL* spectrum at 18 K and $P_{exc}$ = 100 W/cm² from GaN:C annealed at $T_{ann}$ = 800 °C. The dashed lines are calculated using Equation (1) with the same parameters as in Figure 1. The inset zooms in the high-energy part of the spectrum.

The inset to this figure depicts the high-energy part of the spectrum. Resonance Raman scattering lines (R4 and R5) are observed at 3.352 and 3.344 eV at distances from the HeCd laser line (3.8135 eV) multiple of the LO phonon energy in GaN (~92 meV). The donor-bound-exciton (DBE) line at 3.487 eV and the free exciton (FE) line at 3.494 eV have about the same intensity. Two LO phonon replicas of the FE line have a characteristic shape and support the assignment of the FE line [19]. The peak at 3.368 eV is likely an exciton bound to structural defects (the Y4 line at 3.35 eV in strain-free GaN) [2]. A weak peak at 3.277 eV is attributed to the donor–acceptor-pair component of the Mg-related UVL

band [3]. A narrow line at 3.172 eV is the ZPL of the BL$_C$ band. The transformation of the *PL* spectrum with temperature (Figure 3) supports the above attributions.

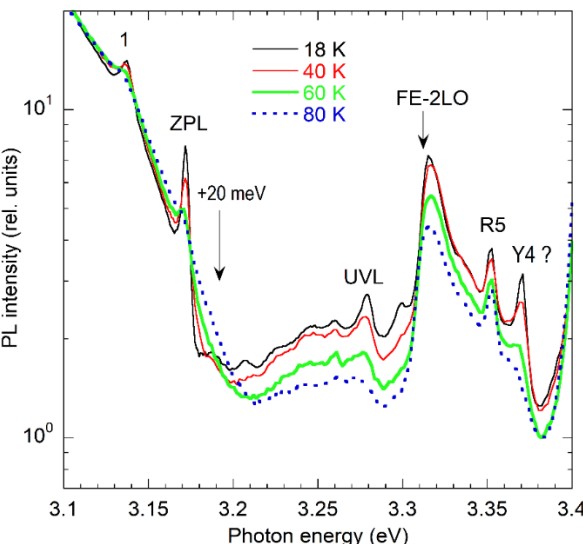

**Figure 3.** Transformation of *PL* spectrum with temperature ($P_{exc}$ = 100 W/cm$^2$). The ZPL of the BL$_C$ band at 3.172 eV decreases, and a shoulder indicated with the arrow "+20 meV" increases with temperature.

From the positions of the BL2 band ZPL at 3.347 meV (before annealing), the DBE line at 3.487 eV, and the FE line at 3.494 eV (before and after annealing), we determine the strain-related blue shift as $\Delta$ = 17 meV and the bandgap $E_g$ = 3.520 eV in the studied GaN:C. The ZPL of the BL$_C$ band is located at 3.172 eV (Figure 3). Assuming that all *PL* lines close to the bandgap are blue-shifted by 17 meV due to strain, we find that $E_0$ = 3.155 eV in unstrained GaN. The ZPL does not shift with excitation intensity, as it is expected for an internal transition from an excited state of a deep donor to the ground state. With increasing temperature, the ZPL intensity decreases.

A shoulder at the high-energy side (indicated with the "+20 meV" arrow in Figure 3) emerges with increasing temperature from 30 to 80 K. The shoulder is attributed to transitions from the conduction band to the ground state of the $C_N^+$. Indeed, the ZPL of the BL$_C$ band, especially its high-energy side, broadens with increasing temperature much stronger than the excitonic lines or than is expected from thermal broadening. The transformation is similar to that for the $C_N H_i$ donor responsible for the BL2 band [7]. By subtracting $E_0$ + 0.02 eV from the bandgap, we obtain the 0/+ transition level of the $C_N$ defect at 0.33 ± 0.01 eV above the valence band in the limit of low temperatures. The ZPL at 3.17 eV and a peak at ~3.14 eV (labeled 1 in Figure 3) were observed in three GaN:C samples annealed at 800 °C.

## 4. Discussion

To study the phonon-related fine structure of the BL$_C$ band, we subtracted the smooth component described with Equation (1) with parameters for the BL$_C$ band shape. The obtained fine structure for two GaN:C samples is shown in Figure 4. A sharp line at 3.17 eV is identified as the ZPL of the BL$_C$ band. The FWHM of this line is about 3 meV when the monochromator slit is small enough. Replicas of the ZPL line at energies multiple of 91.2 meV are the crystal LO phonon replicas with phonon energy $\hbar\Omega_{LO}$. A set of lines with $\hbar\Omega_1$ = 34.3 meV is attributed to a pseudo-local phonon mode.

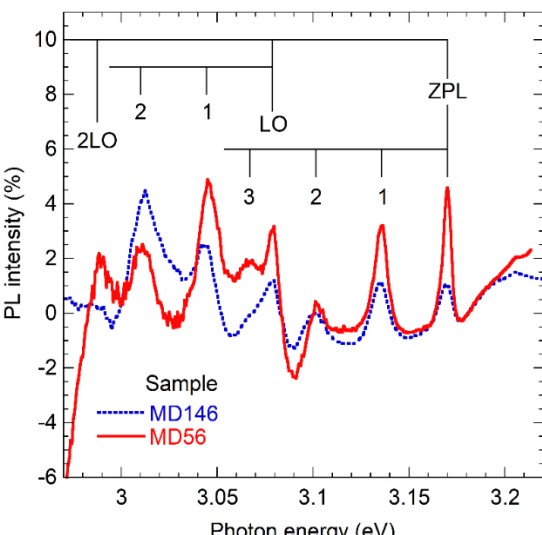

**Figure 4.** *PL* spectra from two GaN:C samples (MD56 and MD146) after subtracting the dependence calculated using Equation (1) with parameters in Figure 1 for the $BL_C$ band. $P_{exc}$ = 100 W/cm$^2$. Two LO phonon replicas of the ZPL line at distances integer of $\hbar\Omega_{LO}$ = 91.2 meV and several pseudo-local phonon replicas (labeled with $n$ = 1, 2 and 3) with $\hbar\Omega_1$ = 34.3 meV are indicated.

Earlier [9], we explained the exponential decay of the $BL_C$ band after a laser pulse, with *PL* lifetime shorter than 1 ns in degenerate *n*-type GaN:C, Si samples and 2.5 ns in high-resistivity GaN:C samples, with the assumption that transitions from an excited state close to the conduction band to the ground 0/+ level and transitions from the conduction band to the same level have similar lifetimes and cannot be resolved. The evolution of the *PL* spectrum with temperature (Figure 3) agrees with the explanation of electron transitions given in Ref. [9] and indicates that the excited state is located at about 0.02 eV below the conduction band.

## 5. Conclusions

We studied carbon-doped GaN annealed at 800 °C. The $C_N$-related $BL_C$ band in these samples revealed the zero-phonon line at 3.17 eV and the phonon-related fine structure caused by LO phonons and a pseudo-local phonon with the energy of 34 meV. The 0/+ transition level of the $C_N$ defect was found at $0.33 \pm 0.01$ eV above the valence band. The $BL_C$ band with a maximum at 2.85 eV was caused by electron transitions from an excited state at about 0.02 eV below the conduction band to the 0/+ level of the $C_N$ defect.

**Funding:** This research was funded by National Science Foundation, grant number DMR-1904861 and by the VCU PeRQ award.

**Data Availability Statement:** The data that support the findings of this study are available from the corresponding author upon reasonable request.

**Acknowledgments:** The author would like to thank Strittmatter (Institute für Physik, Magdeburg, Germany) for providing GaN:C samples and Ye (VCU, USA) for annealing these samples.

**Conflicts of Interest:** The author declares no conflict of interest.

**Sample Availability:** Samples of the compounds are available from the authors.

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
