# Peer review of "Fine Structure of the Carbon-Related Blue Luminescence Band in GaN"

_solids, doi:10.3390/solids3020016_

Round 1
Reviewer 1 Report
The author has studied the photoluminescence property of C-related GaN structure. In general, the proposed work is interesting. However, one of my primary concerns is the plagiarism report, 53 %including references and 42% without references. In that case, I will refer this to the editorial board. If the editorial board still finds this work suitable, please find some of my comments in the pdf attached.

Reviewer 2 Report
1.In the introduction, some other fluorescent materials can be added for comparison, such as carbon quantum dots, graphene quantum dots, etc. References available: Sci. Adv., 2020, 6, eabb6772.ï¼›Molecules, 2021, 26, 4994.
2.A schematic diagram of the material needs to be added, so that the reader can understand this article at a glance.
3.Morphological and structural characterization of materials, such as atomic force microscopy and transmission electron microscopy, needs to be added, so that factors affecting optical properties can be more clearly seen.
Round 2
Reviewer 1 Report
The author has answered all of the queries and significant amendments are made in the revised version of the manuscript. If the journal agrees about the plagiarism issue ( explanation given by the author), from my side this revised version can be accepted for its publication.